# Short- and Long-Term Impact of Smoking Donors in Lung Transplantation: Clinical and Pathological Analysis

**DOI:** 10.3390/jcm10112400

**Published:** 2021-05-28

**Authors:** Marco Schiavon, Andrea Lloret Madrid, Francesca Lunardi, Eleonora Faccioli, Giulia Lorenzoni, Giovanni Maria Comacchio, Alessandro Rebusso, Andrea Dell’Amore, Marco Mammana, Samuele Nicotra, Fausto Braccioni, Dario Gregori, Emanuele Cozzi, Fiorella Calabrese, Federico Rea

**Affiliations:** 1Thoracic Surgery Division, Department of Cardiac, Thoracic, Vascular Sciences and Public Health, Padova University Hospital, 35128 Padova, Italy; marco.schiavon@unipd.it (M.S.); andrea.lloretmadrid@aopd.veneto.it (A.L.M.); eleonora.faccioli@unipd.it (E.F.); giovannimaria.comacchio@aopd.veneto.it (G.M.C.); alessandro.rebusso@aopd.veneto.it (A.R.); marco.mammana@aopd.veneto.it (M.M.); samuele.nicotra@aopd.veneto.it (S.N.); federico.rea@unipd.it (F.R.); 2Pathology Division, Department of Cardiac, Thoracic, Vascular Sciences and Public Health, Padova University Hospital, 35128 Padova, Italy; francesca.lunardi@unipd.it (F.L.); fiorella.calabrese@unipd.it (F.C.); 3Statistics Division, Department of Cardiac, Thoracic and Vascular Sciences and Public Health, Padova University Hospital, 35128 Padova, Italy; giulia.lorenzoni@unipd.it (G.L.); dario.gregori@unipd.it (D.G.); 4Respiratory Pathophysiology Division, Padova University Hospital, 35128 Padova, Italy; fausto.braccioni@aopd.veneto.it; 5Transplant Immunology Unit, Department of Cardiac, Thoracic and Vascular Sciences and Public Health, Padova University Hospital, 35128 Padova, Italy; emanuele.cozzi@unipd.it

**Keywords:** smoking donors, lung, transplantation, marginal donors

## Abstract

Background: The use of smoking donors (SD) is one strategy to increase the organ pool for lung transplantation (LT), but the benefit-to-risk ratio has not been demonstrated. This study aimed to evaluate the impact of SD history on recipient outcomes and graft alterations. Methods: LTs in 293 patients were retrospectively reviewed and divided into non-SD (*n* = 225, group I), SD < 20 pack-years (*n* = 45, group II), and SD ≥ 20 pack-years (*n* = 23, group III) groups. Moreover, several lung donor biopsies before implantation (equally divided between groups) were evaluated, focusing on smoking-related lesions. Correlations were analyzed between all pathological data and smoking exposure, along with other clinical parameters. Results: Among the three groups, donor and recipient characteristics were comparable, except for higher Oto scores and age in group III. Group III showed a longer intensive care unit (ICU) and hospital stay compared with the other two groups. This finding was confirmed when SD history was considered as a continuous variable. However, survival and other mid- and long-term major outcomes were not affected by smoking history. Finally, morphological lesions did not differ between the three groups. Conclusions: In our study, SDs were associated with a longer post-operative course, without affecting graft aspects or mid- and long-term outcomes. A definition of pack-years cut-off for organ refusal should be balanced with the other extended criteria donor factors.

## 1. Introduction

Lung transplantation (LT) is a well-established procedure for several end-stage lung diseases. However, the extensiveness of this procedure is limited, mainly because of a globally diffuse organ shortage. Strategies proven to expand the lung donor pool include optimal intensive-care donor management, donation after cardiac death, and the use of extended criteria donors [1,2,3,4]. The latter strategy has been widely investigated and is considered acceptable, especially for recipients affected by rapidly progressive illnesses [3,5,6,7,8].

Among the extended donor criteria, a positive donor smoking history, generally considered as exposure of more than 20 pack-years, remains controversial, with some evidence of detrimental effects on early recipient outcomes but without a significant impact on long-term survival [9,10,11,12,13,14,15,16]. Based on the literature, the exclusion of heavy smokers from lung donation does not seem justified.

However, to the best of our knowledge, no studies have investigated the effect of donor smoking history using precise pack-years exposure but have rather compared patients in wide clusters with distant pack-year cut-offs. The main goal of this study was to evaluate the impact of donor smoking exposure on lung transplantation outcomes. We also attempted to find a relationship between donor smoking history, microscopic graft damage, and post-transplant outcomes.

## 2. Materials and Methods

### 2.1. Study Population

In this single-center retrospective study, we reviewed a cohort of patients who underwent LT in our institution between May 1995 and June 2015. During this period, we performed 355 LTs (110 single, 243 bilateral, and 2 heart-lung transplantations). The main clinical information about the donor, the recipient, and the surgical procedure was collected. Donor data comprised the main clinical and functional characteristics, including Oto score [17]. Donor smoking history was reported as pack-years (number of packs of cigarettes in a day multiplied by the number of smoking years).

Due to a lack of precise smoking history information, we excluded 62 patients from the study, thus leaving 293 patients for further analysis. The institutional review board gave approval for this study (4539/AO/18).

Preoperative recipient data included age, sex, weight, height, body mass index (BMI), native disease, and, eventually, emergency status (pre-transplant mechanical circulatory or ventilatory support). Surgical procedure parameters included ischemic time (median time of both lungs for bilateral procedures), operative room (OR) time, extracorporeal support, and type of transplantation (single, bilateral, or heart-lung).

Post-operative data included the main early outcomes of ventilation time; intensive care unit (ICU) and hospital stay; and primary graft dysfunction (PGD) graded at 0, 24, 48, and 72 h [18]. Thirty-day and in-hospital mortality and occurrence of early (up to 90 days post-transplant) complications (excluding other reported outcomes) were also recorded. In addition, the following mid- and long-terms outcomes were evaluated: late complications (starting from the third postoperative month); infections; acute rejection [19]; chronic rejection [20]; peak FEV1; and 6-, 12-, and 24-month FEV1. Long-term survival analysis was based on the last follow-up update (December 2020) or retransplantation/death date.

### 2.2. Pathological Analysis

Donor lung pathological examinations were performed in 97 cases, equally distributed between the three groups, and were available in cases of downsizing lobar resection during transplantation or single LT after bilateral harvesting. Biopsy tissue samples performed in apparently normal lung parenchyma were formalin-fixed, paraffin-embedded, and stained with hematoxylin-eosin. Different smoking-related lesions (emphysema, anthracosis, bronchiolitis, and interstitial inflammation) were evaluated by a dedicated thoracic pathologist using a scoring system from 0 to 3 (0 = absence of lesion, 1 = involvement of less than 30% of parenchyma, 2 = involvement of 30–60% of parenchyma, and 3 = involvement of more than 60% of parenchyma). The 3-year acute cellular rejection (ACR) index was quantified as the number of transbronchial biopsies (TBBs) with ACR ≥ 1/total number of TBBs × 100, using TBBs performed in the first three years post-transplantation. Similarly, the ACR ≥ A2 index (number of TBBs with ACR ≥ A2/total number of TBBs × 100) was also considered. The presence of airway inflammation was determined based on evidence of lymphocytic bronchiolitis (LB) in all biopsy specimens.

### 2.3. Statistical Analysis

Descriptive statistics were reported as quartile I/median/quartile III for continuous variables and as percentages (absolute numbers) for categorical variables. Wilcoxon-Kruskal-Wallis and Pearson’s Chi-square tests were performed to compare the distribution of continuous and categorical variables, respectively.

A regression approach, both univariable and multivariable, was employed to assess the effect of recipients’ smoking history on clinical outcomes of interest. A logistic regression approach was adopted for binary outcomes. Results were reported as odds ratio (OR), 95% confidence interval (CI), and *p*-value. An ordinal regression approach was employed for nominal outcomes. Results were reported as OR, 95% CI, and *p*-value. A Gamma model was employed for continuous outcomes, given the non-normal distribution of all continuous outcomes considered. Results were reported as marginal effect, 95% CI, and *p*-value. The marginal effect was computed from the partial derivatives of the marginal expectation.

Analyses were performed using R software (R Foundation, Vienna, Austria) [21] with rms [22] and MASS packages.

## 3. Results

### 3.1. Three-Group Analysis

Among the 293 patients considered in the study, 225 recipients had no positive donor smoking history (group I), 45 had a donor smoking history between 1 and 20 pack-years (group II), and 23 had a donor smoking history of 20 pack-years or more (group III).

Table 1 summarizes the most important baseline recipient and donor characteristics. As shown, recipients were similar in all pre- and intra-operative parameters. Donor characteristics were comparable between the three groups in many aspects, except for age: donors in group III were significantly older than those in groups I and II (47 vs. 38 and 35 years, *p* < 0.001). In addition, as expected, group III showed a higher Oto score compared with groups I and II (4.5 vs. 2 and 2, *p* < 0.001).

Table 2 summarizes early- and mid-term clinical outcomes after transplantation. A significant increase in median ventilation time (36 vs. 42 vs. 58 h, *p* = 0.04) and ICU stay (6 vs. 7 vs. 17 days, *p* = 0.003) was observed with increasing smoking history. At the same time, group III also presented a longer duration of post-transplant hospitalization and reduced respiratory function compared with the other two groups, although these differences were not statistically significant. While the ACR index was similar among groups (*p* = 0.165), the probability of grade A2 or worse decreased significantly as donor smoking history increased (37% vs. 20% vs. 13%, *p* = 0.009).

Finally, chronic rejection prevalence was not different between groups, nor were PGD rate and short-term mortality (both in-hospital and 30-day mortality).

Multivariate analysis (Table 3) confirmed the detrimental effect of high donor smoking on recipient ICU stay (Effect 9.7, 95% CI 1.9;18.6, *p* = 0.02) but not on ventilation time. 

At the same time, this analysis found a correlation between group III and longer hospital stay (Effect 15, 95% CI 3.8;27, *p* = 0.01).

Finally, an association between recipient female sex and PGD rate at 72 h and in-hospital mortality (Effect 2.3, 95% CI 1.1;4.5, *p* = 0.02 and Effect 2.6, 95% CI 1.1;6.1, *p* = 0.03, respectively) was observed.

Table 4 shows the multivariable analysis of post-discharge outcomes. No influence of donor smoking history was observed on any parameter evaluated. On the contrary, older recipients seem less likely to develop ACR (Effect 1.0, 95% CI 0.9;1.0, *p* = 0.06), especially when considering only ≥A2 grade cases (Effect 0.9, 95% CI 0.9;1.0, *p* = 0.02). Concerning respiratory performance, a worse 12-month FEV1 % was observed in female recipients (Effect −0.6, 95% CI −0.9;−0.2, *p* = 0.002) and in patients with restrictive diseases and specifically idiopathic pulmonary fibrosis patients (Effect −0.6, 95% CI −1.1;−0.2, *p* = 0.01 and Effect −0.5; 95% CI −0.9;−0.1, *p* = 0.002, respectively).

Regarding long-term outcomes, overall 5-year survival (Figure 1) was similar in groups I and II (42.5% vs. 41.5%), and was slightly higher, although not significantly, in these two groups than in group III (23.3%, *p* = 0.44).

### 3.2. Continuous Variable Analysis

Univariate analysis (Table 5) showed a significant increase in postoperative ventilation time (*p* = 0.04) as well as ICU and hospital stay (*p* = 0.04 and *p* = 0.03, respectively) as donor smoking history increased. No effect was observed on the other outcomes evaluated, particularly on 12-month FEV1.

Multivariate analysis (Table 6) confirmed the donor smoking history impact on ICU (Effect 0.2, 95% CI 0.0001;0.5, *p* = 0.04) and hospital stay (Effect 0.3, 95% CI 0.01;0.6, *p* = 0.03) but not on ventilation time.

### 3.3. Pathological Analysis

Considering donor smoking history as a continuous variable for multivariate analysis (Table 7), no evidence of a donor smoking history impact on graft pathology findings were observed. On the contrary, a direct effect of donor age on the degree of emphysema (OR 1.1, 95% CI 1.0;1.1, *p* = 0.03) and anthracosis (OR 1.1, 95% CI 1.0;1.1, *p* = 0.001) was observed.

## 4. Discussion

The use of extended criteria organs is one of the most employed and extensively investigated strategies to overcome the lung donor shortage and expand lung transplantation activity. Despite some minor evidence of worse outcomes [23], liberalization of donor criteria is considered acceptable to satisfy organ demand, particularly for patients suffering from rapidly progressive lung diseases [7,24]. Concerning donor smoking history of >20 pack-years, probably one of the most debated marginal donor criteria, previously published studies reported a negative impact only on the early post-transplant period, while long-term survival seemed uncompromised [9,11,13,15,16,17,18,19,20,21,22,23,24,25]. Our findings confirmed the impact of donor smoking history on several early recipients’ post-operative outcomes (ICU and hospital stay), but without any impact on ventilation time and PGD rate as described by other authors [9,13,16,25].

We do not have a precise explanation for the increase in ICU stay of patients transplanted with donor organs with a history of smoking, considering similar ventilation time and PGD among the three groups. Nevertheless, this may be related to the older age of both the recipients and the donors in group III, which could have led to careful and protective anesthesiologic management in these patients.

Long-term results showed no differences among the three groups in survival rate, early and late complications (bronchial or immunologic), and pulmonary function. Multivariate analysis did not reveal a correlation between the donor’s smoking history and lung function; however, compared with groups II and I, recipients of group III showed a 21.5% and 15% lower FEV1 at 6 months post-LT, a 10% and 14% lower FEV1 at 12 months, and a 14% and 27% lower FEV1 at 24 months, respectively.

This fact may be related to the presence of significantly older donors in the group of patients showing lower respiratory performance, a factor known to impact recipient respiratory function tests [26]. A common problem in studies investigating the relationship between donor smoking and lung transplantation outcomes is the marked disproportion between sample sizes, with very small samples of donors with positive smoking histories, particularly for the heavy smoking category. This intrinsic statistical problem limits the clinical interpretation of any evidence that emerges, especially for understanding the levels of smoking beyond which transplant outcomes are markedly poorer. In order to overcome this issue, we adopted an innovative approach in our study, which consisted of considering donor smoking exposition as a continuous variable using a precise computation of the pack-years history of every donor included. The first result that emerged from this analysis was the linear (not exponential) effect of cigarette smoking history on two main early post-transplant outcomes (ICU and hospital stay). This finding implies two main observations: first, given the linear fashion of the effect, it seems impossible to determine a cut-off beyond which recipient adverse outcomes dramatically increase; second, the effect is already present at minimal pack-year levels, well below the traditional 20-pack-year level, meaning that even a very light donor smoking history has an effect. This is surprising, given that ideal donors have always included smokers below 20 pack-years.

Bearing in mind these considerations, we could define donor smoking history as a non-modifiable risk factor that needs to be considered at the time of lung offer. As a single criterion, donor smoking history does not appear sufficient for organ refusal, and it should be weighed with other extended criteria when present as well as other recipient risk factors (such as primary diagnosis, clinical severity, and age).

Obviously, to carefully evaluate a lung donor as suitable, the evaluation of a CT scan is mandatory before retrieval to exclude the presence of lung nodules, parenchymal thickening, anatomical abnormalities, atelectasis, and emphysema.

In our center, the donor’s evaluation is always performed by a senior surgeon and, especially in the case of smoking history, the visual evaluation of the CT scan is important to quantify emphysema. Unfortunately, in our study, a CT scan was not always available before retrieval and, for this reason, the quantification of emphysema was done directly at the retrieval time through the visual inspection, palpation, and recruitment maneuvers of the lungs.

In our opinion, there is no direct correlation between pack-years and the extension of emphysema. We retrieved lungs from 23 donors with a smoking history ≥ 20 pack-years, considering them suitable; this can be explained by the fact that emphysema is a multifactorial event not only caused by smoking.

Thus, a correct donor-recipient match seems vital, as demonstrated by Pierre et al. [6], who described a 5.3% 30-day mortality when matching a high-risk recipient and an ideal donor, a 15% mortality rate with an ideal recipient and a marginal donor, and a 22.2% mortality rate with a high-risk recipient and a marginal donor.

Another peculiarity of our study is the investigation of the correlation between donor smoking history and allograft microscopic alteration at the time of transplantation. Different from other work [27] that analyzed only the main airway or donor bronchoalveolar lavage (BAL), we were able to examine a large tissue sample. Based on our limited case series, it seems that smoking rate does not significantly impact morphologic alterations. We only observed a correlation between donor age, emphysema, and anthracosis grade. Similarly, Sabashnikov et al. [16] found a higher prevalence of inflammation and metaplasia in bronchial specimens from heavy smokers, but the difference was not significant.

Outside the field of lung transplantation, different studies have documented the growing prevalence of microscopic alterations with the increase of smoking history, but they do not completely fit our field of investigation since they are based on populations that are very different from the lung donor population; the former individuals are older, have a much higher smoking exposure, and often have associated pulmonary diseases, even neoplastic ones [28,29,30,31,32].

Our study was limited by several factors, mostly related to the asymmetric distribution of donor smoking history, with a small population sample in the heavy-smoker donor group. This fact, in addition to post-transplant mortality, very likely affected long-term outcomes as few heavy-smoker donors could be evaluated.

In this light, a multi-center study would be desirable. In addition, the study period covering approximately 20 years of activity certainly involved limitations for the evolution of both pharmacological and anesthetic treatments.

## 5. Conclusions

In conclusion, in our experience, the use of donors with a positive smoking history is associated with a longer post-operative course proportional to pack-years, without any influence on other major early- and long-term results or on pulmonary tissue preservation. Thus, in our opinion, the decision to accept this type of graft should consider not only the number of pack-years but also other donor and/or recipient risk factors, along with the donor shortage situation in the country and waitlist overcrowding.

## Figures and Tables

**Figure 1 jcm-10-02400-f001:**
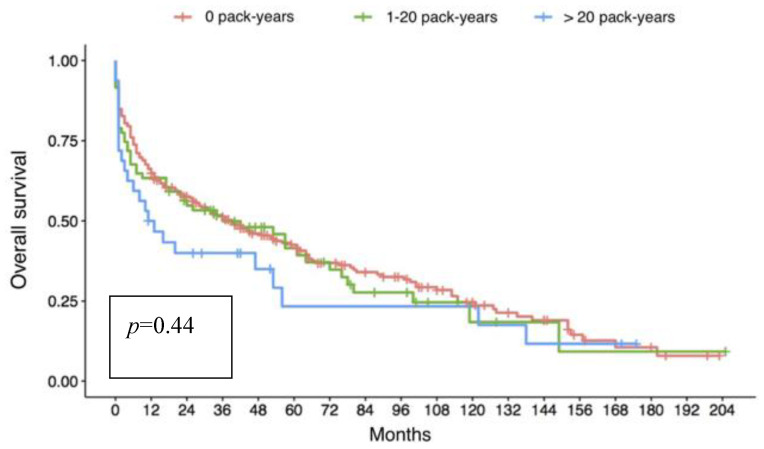
Kaplan-Meyer analysis of overall survival of the three groups according to donor smoking history; no difference can be observed between the three populations even though a slight trend of reduced survival is present in group III (donors with a smoking history of more than 20 pack-years).

**Table 1 jcm-10-02400-t001:** Baseline recipient and donor characteristics ^1^.

GROUP	0 Pack-Years (*N* = 225)	<20 Pack-Years (*N* = 45)	>20 Pack-Years (*N* = 23)	*p*
Recipients				
Age (years)	49 (34–58)	45 (32–57)	55 (43–61)	0.28
Female sex	81 (36%)	23 (51%)	10 (43%)	0.15
Weight (kg)	64 (50–76)	60 (52–75)	71 (51–82)	0.47
Height (cm)	160 (160–174)	164 (158–170)	167 (159.5–175)	0.21
BMI	22 (19–26)	23 (19–26)	25 (20–28)	0.29
Type of transplant				0.43
Bilateral	148 (66%)	34 (76%)	15 (65%)	
Single	77 (34%)	11 (24%)	8 (35%)	
Circulatory support during LT	63 (28%)	18 (40%)	6 (26%)	0.25
Disease				0.99
IPF	74 (33%)	15 (33%)	8 (35%)	
COPD	33 (15%)	7 (16%)	4 (17%)	
CF	61 (27%)	13 (29%)	6 (26%)	
Bronchiectasis	14 (6%)	3 (7%)	1 (4%)	
IPAH	3 (1%)	0 (0%)	1 (4%)	
Other	40 (18%)	7 (16%)	3 (13%)	
Emergency status	14 (6%)	7 (16%)	2 (9%)	0.10
Donors				
Age (years)	35 (21–49)	38 (29–48)	47 (44.5–52)	<0.01
Female sex	91 (40%)	24 (53%)	12 (52%)	0.19
Weight (kg)	70 (60–80)	65 (55–75)	68 (58–75)	0.39
Height (cm)	170 (165–180)	168 (165–176)	166 (160–175)	0.11
BMI	23.4 (21.2–25.5)	23.1 (21.1–25.9)	24.5 (22.7–25)	0.54
Cause of death				
Head trauma	102 (45%)	11 (24%)	4 (17%)	
Stroke	97 (43%)	28 (62%)	17 (74%)	
Other	40 (18%)	6 (12%)	2 (8%)	
Ventilation time (hours)	55 (36–98)	48 (33–69)	48 (39–80)	
ICU stay (hours)	59 (32–96)	52 (35–72)	48 (29–96)	
PaO_2_ with FiO_2_ 100% (mmHg)	470 (409–521)	445 (396–499)	439 (379–466)	
Oto Score	2 (1–4)	2 (1–4)	4.5 (3–6)	
Mean ischemic time (minutes)	354 (271–398)	328 (265–399)	385 (322–403)	

^1^ Data are presented as median (interquartile range) values or absolute frequency (percent), according to the variable nature. BMI, body mass index; LT, lung transplantation; IPF, idiopathic pulmonary fibrosis; COPD, chronic obstructive pulmonary disease; CF, cystic fibrosis; IPAH, idiopathic pulmonary arterial hypertension; ICU, intensive care unit.

**Table 2 jcm-10-02400-t002:** Early- and mid-term post-operative outcomes ^1^.

GROUP	0 pack-years (*N* = 225)	<20 Pack-Years (*N* = 45)	>20 Pack-Years (*N* = 23)	*p*
Ventilation time (hours)	36 (15–72)	42 (29–155)	58 (23–385)	0.04
ICU stay (days)	6 (4–14)	7 (5–19)	17 (7–39)	0.003
Hospital stay (days)	30 (25–40)	30 (24–45)	33 (25–50)	0.38
PGD at 0 h				
1	21 (10%)	2 (5%)	1(5%)	0.53
2	53 (25%)	8 (18%)	7 (32%)	
3	38 (18%)	10 (23%)	6 (27%)	
PGD at 24 h				
1	20 (10%)	5 (11%)	0 (0%)	0.60
2	54 (26%)	8 (18%)	7 (32%)	
3	30 (14%)	5 (11%)	6 (27%)	
PGD at 48 h				
1	19 (9%)	3 (7%)	3 (14%)	0.37
2	52 (25%)	9 (20%)	6 (27%)	
3	32 (15%)	4 (9%)	0 (0%)	
PGD at 72 h				
1	13 (6%)	1 (2%)	1 (5%)	0.76
2	67 (32%)	12 (28%)	9 (41%)	
3	25 (12%)	4 (9%)	3 (14%)	
Acute rejection	139 (62%)	24 (53%)	10 (43%)	0.17
Acute rejection ≥ A2	84 (37%)	9 (20%)	3 (13%)	0.009
Chronic rejection	56 (25%)	6 (13%)	5 (22%)	0.24
In-hospital mortality	40 (18%)	12 (27%)	7 (30%)	0.18
30-day mortality	31 (14%)	10 (22%)	5 (22%)	0.26
FEV1 % at 6 months	72 (56–84.5)	78.5 (60–91)	57 (49–73.5)	0.15
FEV1 % at 12 months	76 (59–90)	72 (55.5–90)	62 (53–75.5)	0.26
FEV1 % at 24 months	79 (60–94)	76 (58–92)	52 (45.5–80)	0.41
Peak FEV1 %	84 (64–100)	88.5 (69–98.5)	74 (65–95)	0.68

^1^ Data are presented as median (interquartile range) values or absolute frequency (percent), according to variable nature; ICU, intensive care unit; PGD, primary graft dysfunction; FEV1, forced expiratory volume in 1 s.

**Table 3 jcm-10-02400-t003:** Multivariate analysis of variables associated with early post-transplant outcomes ^1^.

	PGD 72	Ventilation Time	ICU Stay	Hospital Stay	In-Hospital Mortality
	OR (95% CI); *p*	Marginal Effect (95% CI); *p*	Marginal Effect (95% CI); *p*	Marginal Effect (95% CI); *p*	OR (95% CI); *p*
Smoking history					
0	Ref	Ref	Ref	Ref	Ref
<20	0.5 (0.3;1.1); 0.1	42.3 (94.8;204.6); 0.56	1.4 (−4.6;7.8); 0.64	1.3 (−6.9;9.9); 0.77	1.30 (0.55;3.08); 0.55
≥20	0.9 (0.3;2.3); 0.76	103.4 (83.6;337.7); 0.31	9.7 (1.9;18.6); 0.02	15 (3.8;27); 0.01	1.13 (0.38;3.39); 0.83
Oto score	1.1 (1.0;1.3); 0.18	10.6 (17.3; 40.7); 0.43	0.5 (−0.6;1.6); 0.36	−0.8 (−2.4;0.8); 0.29	1.2 (1.0;1.4); 0.06
Ischemic time	1.0 (1.0;1.0); 0.10	−0.1 (-0.6;0.4); 0.74	0.002 (−0.02;0.02); 0.87	0.01 (0.02;0.04); 0.53	1.0 (1.0;1.0); 0.57
Donor age	1.0 (1.0;1.0); 0.18	0.8 (−3.8;5.3); 0.69	0.2 (−0.01;0.4); 0.05	0.1 (−0.2;0.3); 0.64	1.0 (0.9;1.1); 0.21
Donor F sex	0.6 (0.3;1.2); 0.12	82.5 (48.3;215.3); 0.22	3.6 (−1.9;9.2); 0.20	9.4 (1.3;17.5); 0.02	0.6 (0.3;1.5); 0.29
Recipient F sex	2.3 (1.1;4.5); 0.02	62.2 (−64.7;192.8); 0.36	5.2 (−0.2;0.7); 0.07	2.3 (−5.7;10.2); 0.57	2.58 (1.1;6.1); 0.03
Recipient age	1.0 (1.0;1.0); 0.42	1.2 (−3.3;5.5); 0.54	0.001 (−0.2;0.2); 0.99	0.2 (−0.1;0.5); 0.17	1.0 (0.9;1.0); 0.79
Disease					
Septic diseases *	Ref		Ref	Ref	Ref
IPF	1.7 (0.7;4.0); 0.24		4.6 (-2.7;11.7); 0.20	1.6 (-8.3;11.4); 0.75	2.0 (0.7;5.7); 0.21
COPD	0.9 (0.4;2.6); 0.93		6.3 (-1.6;14.4); 0.11	8.0 (-3.1;19)	1.6 (0.5;5.2); 0.46
Other interstitial disease	1.9 (0.7;4.8); 0.19		0.9 (-6.4;8.8); 0.81	0.9 (0.3;3.2); 0.88	0.9 (0.3;3.2); 0.88
Other	1.8 (0.6;5.4); 0.29		16.1 (7.1;25.9); 0.01	4.4 (1.4;13.7); 0.01	4.4 (1.4;13.7); 0.01
	**PGD 72**	**Ventilation Time**	**ICU Stay**	**Hospital Stay**	**In-Hospital Mortality**
	**OR (95% CI); *p***	**Marginal Effect (95% CI); *p***	**Marginal Effect (95% CI); *p***	**Marginal Effect (95% CI); *p***	**OR (95% CI); *p***
Smoking history					
0	Ref	Ref	Ref	Ref	Ref
<20	0.5 (0.3;1.1); 0.1	42.3 (94.8;204.6); 0.56	1.4 (−4.6;7.8); 0.64	1.3 (−6.9;9.9); 0.77	1.30 (0.55;3.08); 0.55
≥20	0.9 (0.3;2.3); 0.76	103.4 (83.6;337.7); 0.31	9.7 (1.9;18.6); 0.02	15 (3.8;27); 0.01	1.13 (0.38;3.39); 0.83
Oto score	1.1 (1.0;1.3); 0.18	10.6 (17.3; 40.7); 0.43	0.5 (−0.6;1.6); 0.36	−0.8 (−2.4;0.8); 0.29	1.2 (1.0;1.4); 0.06
Ischemic time	1.0 (1.0;1.0); 0.10	−0.1 (-0.6;0.4); 0.74	0.002 (−0.02;0.02); 0.87	0.01 (0.02;0.04); 0.53	1.0 (1.0;1.0); 0.57
Donor age	1.0 (1.0;1.0); 0.18	0.8 (−3.8;5.3); 0.69	0.2 (−0.01;0.4); 0.05	0.1 (−0.2;0.3); 0.64	1.0 (0.9;1.1); 0.21
Donor F sex	0.6 (0.3;1.2); 0.12	82.5 (48.3;215.3); 0.22	3.6 (−1.9;9.2); 0.20	9.4 (1.3;17.5); 0.02	0.6 (0.3;1.5); 0.29
Recipient F sex	2.3 (1.1;4.5); 0.02	62.2 (−64.7;192.8); 0.36	5.2 (−0.2;0.7); 0.07	2.3 (−5.7;10.2); 0.57	2.58 (1.1;6.1); 0.03
Recipient age	1.0 (1.0;1.0); 0.42	1.2 (−3.3;5.5); 0.54	0.001 (−0.2;0.2); 0.99	0.2 (−0.1;0.5); 0.17	1.0 (0.9;1.0); 0.79
Disease					
Septic diseases *	Ref		Ref	Ref	Ref
IPF	1.7 (0.7;4.0); 0.24		4.6 (−2.7;11.7); 0.20	1.6 (−8.3;11.4); 0.75	2.0 (0.7;5.7); 0.21
COPD	0.9 (0.4;2.6); 0.93		6.3 (−1.6;14.4); 0.11	8.0 (−3.1;19)	1.6 (0.5;5.2); 0.46
Other interstitial disease	1.9 (0.7;4.8); 0.19		0.9 (−6.4;8.8); 0.81	0.9 (0.3;3.2); 0.88	0.9 (0.3;3.2); 0.88
Other	1.8 (0.6;5.4); 0.29		16.1 (7.1;25.9); 0.01	4.4 (1.4;13.7); 0.01	4.4 (1.4;13.7); 0.01

* indicates reference variable for comparison. ^1^ 95% CI, 95% confidence interval; COPD, chronic obstructive pulmonary disease; F, female; ICU, intensive care unit; IPF, idiopathic pulmonary fibrosis; PGD, primary graft failure.

**Table 4 jcm-10-02400-t004:** Multivariate analysis of variables associated with mid and long-term post-transplant outcomes ^1^.

	Acute Rejection OR (95% CI); *p*	Acute Rejection ≥ A2 OR (95% CI); *p*	Chronic Rejection OR (95% CI); *p*	12 Months FEV1 Marginal Effect (95% CI); *p*
Smoking history				
0	Ref	Ref	Ref	Ref
<20	0.8 (0.4;1.6); 0.46	0.7 (0.4;1.4); 0.35	0.5 (0.2;1.3); 0.14	0.01 (−0.3;0.3); 0.93
≥20	0.7 (0.3;1.9); 0.52	0.9 (0.4;2.5); 0.89	1.13 (0.4;3.5); 0.84	−0.3 (−0.9;0.3); 0.27
Oto Score	0.9 (0.8;1.1); 0.24	0.9 (0.8;1.1); 0.22	1.0 (0.8;1.1); 0.80	0.0001 (−0.1;0.1); 1.00
Ischemic time	1.0 (1.0;1.0); 0.24	1 (1.0;1.0); 0.66	1.0 (1.0;1.0); 0.24	−0.001 (−0.003;0.0001); 0.07
Donor age	1.0 (1.0;1.0); 0.43	1.0 (0.9;1.0); 0.17	1.0 (1.0;1.0); 0.08	−0.01 (−0.01;0.01); 0.33
Donor F sex	1.1 (0.6;2.2); 0.73	1.2 (0.6;2.4); 0.54	0.9 (0.4;1.9); 0.68	−0.03 (−0.4;0.3); 0.89
Recipient F sex	0.7 (0.3;1.3); 0.21	0.6 (0.3;1.3); 0.20	1.3 (0.6;2.8); 0.54	−0.54 (−0.9;−0.2); 0.002
Recipient age	1.0 (0.9;1.0); 0.06	0.9 (0.9;1.0); 0.02	1.0 (1.0;1.0); 0.66	−0.004 (−0.02;0.01); 0.52
Disease				
Septic diseases *	Ref	Ref	Ref	Ref
IPF	0.9 (0.4;2.0); 0.75	0.9 (0.4;2.1); 0.84	1.4 (0.5;3.8); 0.50	−0.6 (−1.1;−0.2); 0.002
COPD	0.5 (0.2;21.3); 0.16	0.5 (0.2;1.2); 0.11	1.9 (0.7;5.3); 0.25	0.4 (−1.1;−0.2); 0.10
Other interstitial disease	0.5 (0.2;1.2); 0.12	0.5 (0.2;1.3); 0.15	2.3 (0.9;6.0); 0.09	−0.5 (−0.9;−0.1); 0.01
Other	0.2 (0.1;0.5); 0.001	0.2 (0.1;0.6); 0.004	1.2 (0.4;4.0); 0.73	−0.2 (−0.7;0.4); 0.55

* indicates reference variable for comparison. ^1^ 95% CI, 95% confidence interval; FEV1, forced expiratory volume in 1 second; IPF, idiopathic pulmonary fibrosis; COPD, chronic obstructive pulmonary disease; F, female.

**Table 5 jcm-10-02400-t005:** Univariate analysis of smoking history association with postoperative outcomes. Donor smoking history is considered a continuous variable.

Outcomes	Odds Ratio (95% CI)	*p*
In-hospital mortality	1.0 (1.0;1.1)	0.13
PGD 72 hours	1.0 (1.0;1.1)	0.26
Ventilation time	5.2 (0.5;11.5)	0.04
ICU stay	0.3 (0.1;0.6)	0.007
Hospital stay	0.3 (0.04;0.7)	0.03
Acute rejection	1.0 (0.9;1.0)	0.12
Acute rejection ≥ A2	1.0 (0.9;1.0)	0.21
Chronic rejection	1.0 (1.0;1.0)	0.59
FEV1 12 months	−0.01 (−0.03;0.00001)	0.44

**Table 6 jcm-10-02400-t006:** Multivariate analysis of variables associated with postoperative ventilation, ICU, and hospital stay. Donor smoking history is considered as a continuous variable ^1^.

	Ventilation Time		ICU Stay		Hospital Stay	
	Effect (95% CI)	*p*	Effect (95% CI)	*p*	Effect (95% CI)	*p*
Smoking history	3.2 (−1.9;9.5)	0.23	0.2 (0.0001;0.5)	0.04	0.3 (0.01;0.6)	0.04
Oto score	11.0 (−16.5;40.7)	0.40	0.5 (−0.6;1.7)	0.43	−0.8 (−2.3;0.8)	0.33
Mean graft ischemic time	−0.1 (−0.6;0.4)	0.73	0.002 (−0.02;0.03)	0.63	0.01 (−0.02;0.04)	0.49
Donor age	0.8 (−3.8;5.3)	0.70	0.2 (0.0006;0.4)	0.92	0.07 (−0.2;0.3)	0.59
Donor F sex	82.3 (−48.2;214.4)	0.22	3.3 (−2.3;9.0)	0.17	0.1 (0.8;17.5)	0.02
Recipient F sex	59.8 (−65.8;188.1)	0.38	5.3 (−0.2;10.9)	0.11	2.4 (−5.8;10.6)	0.55
Recipient age	1.2 (−3.3;5.4)	0.54	0.007 (−0.2;0.2)	0.95	0.2 (−0.1;0.5)	0.17
Disease						
Septic diseases *			Ref		Ref	
IPF			4.8 (−2.6;12.0)	0.19	2.0 (−8.1;12.0)	0.70
COPD			6.1 (−1.8;14.2)	0.13	7.8 (−3.4;19.2)	0.17
Other interstitial diseases			1.1 (−6.3;9.1)	0.78	0.4 (−10.0;11.3)	0.94
Other			16.2 (7.1;26.2)	0.005	13.1 (0.7;26.4)	0.05

* Reference variable for comparison. ^1^ 95% CI, 95% confidence interval; ICU, intensive care unit; BMI, body mass index; IPF, idiopathic pulmonary fibrosis; COPD, chronic obstructive pulmonary disease; F, female.

**Table 7 jcm-10-02400-t007:** Multivariate analysis of variables associated with graft pathology ^1^.

	Emphysema OR (95% CI)	*p*	Bronchiolitis OR (95% CI)	*p*	Anthracosis OR (95% CI)	*p*	Interstitial Inflammation OR (95% CI)	*p*
Donor smoking history	1.0 (0.9;1.0)	0.86	1.0 (0.9;1.1)	0.38	1.0 (1.0;1.1)	0.55	1.0 (0.9;1.0)	0.47
Donor age	1.1 (1.0;1.1)	0.03	1.0 (0.9;1.0)	0.79	1.1 (1.0;1.1)	0.001	1.0 (1.0;1.1)	0.09
Ischemic time	1.0 (0.0;1.0)	0.86	1.0 (1.0;1.0)	0.26	1.0 (1.0;1.0)	0.24	1.0 (1.0;1.1)	0.36
Intubation time	1.0 (1.0;1.0)	0.34	1.0 (1.0;1.0)	0.45	1.0 (1.0;1.0)	0.36	1.0 (1.0;1.0)	0.92

^1^ 95% CI, 95% confidence interval.

## Data Availability

The data presented in this study are available on request from the corresponding author. The data are not publicly available to protect the privacy of the patients.

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
