# Peer review of "Short- and Long-Term Impact of Smoking Donors in Lung Transplantation: Clinical and Pathological Analysis"

_jcm, 2021, doi:10.3390/jcm10112400_

Round 1

Reviewer 1 Report

The authors describe the impact of heavy smoking on lung function after Lung transplantation. They found a significant correlation between length of active smoking and a prolonged postoperative ventilation time and ICU stay. 

The manuscript is precise and presented in an adequate manner.

However, in line 84 parenchima should be replaced by parenchyma. 

Author Response

Response to Reviewer 1 Comments

Point 1: The authors describe the impact of heavy smoking on lung function after Lung transplantation. They found a significant correlation between length of active smoking and a prolonged postoperative ventilation time and ICU stay. The manuscript is precise and presented in an adequate manner. However, in line 84 parenchima should be replaced by parenchyma. 

Response 1: Dear reviewer, we really appreciate your comment on the manuscript. We have changed, as suggested, the word "parenchima" with the correct form "parenchyma" in line 84. 

Reviewer 2 Report

This study on smoking history confirms previous studies that smoking history should not be a strong exclusion criteria in dicision making to transplant.

I have some major comments :

1/ smoking history is not emphysema and yet we somehow make it the same.

Respect for having this accurate packyear info as we were not able to have this in our center as asking this a the time of donation is difficult as it is a very emotional moment.

But again smoking history is only a direction for looking at smoking related emphysema.

Can you score emphysema somehow on CT scans ? is there a pre or post transplant moment where emphysema could be scored ?

Not all smoker get COPD so this again is a bridge to far I would say.

If this scoring is possible can it be related to the pack years ? (I hypothesis they do NOT correlate as this is obvious).

2/ smoking or pack years in a way is not the exclusion but the surgeon going for prelavation will during examination look (or feel) more for the emphysema and see if the deflate slower, bullae presence and so on.

I know form our center that rejected lungs for emphysema can look very bad. And also that soem donor lungs used hold more pronounced emphysema get their pack years was not extraordinary.

This discripance can you study this ? is their info from surgeon where they describe what they see and feel ?

3/ the biopsy study is very nice yet and biopsy does not reflect the emphysema and it is never homogenious. Could you quantify emphysema ? and also could you quantify the inflammation around the airways ?

We know that donor exposure to pollution of smoking give inflammation of the small airway and that this impact on short term outcome, which you see. Can you link this ?

LM would be cool the quantify and relate to the CT emphysema and FEV1 at 1 month and best FEV1.

4/ your FEV1 data show that the 20PY group have a lower FEV1 early on post LTx but that the other group catch up. Do you have 1 month data ?

May it be that the early lower lung function results in the 20PY group makes that they have no reserve in the beginning causing the complication and longer ICU stay and hospital admission ?

5/ the outcome in the long term is not it seem. But are you misleaded (as all previous studies). The common feeling was that a lung from a smoker is inferior and could only be given to a COPD patients (as he does not deserve a top quality lung). Yet COPD patients in general do better in terms of survival. So could this bias the findings and be misleading ?

If you could tackle those question you would give important information for surgeons and pulmonologist in donor acceptance making and selecting of receptors even. And why not giving better tool for exluding donor acceptance due to emphysema. I personnally have seem lungs accepted with emphysema which was very pronounced and also donor rejection for smoking but the emfysema was limited…So very relavent study but take the real challenge.

But again very nice work and respect for having this nice data on smoking history.

Author Response

Response to Reviewer 2 Comments

Dear reviewer, we really thank you for your useful comments. We added a paragraph in the discussion (lines 243-255) trying to answer your main issues. Please see our replies to your comments below:

Point 1: Smoking history is not emphysema and yet we somehow make it the same. Respect for having this accurate pack-year info as we were not able to have this in our center as asking this a the time of donation is difficult as it is a very emotional moment. But again smoking history is only a direction for looking at smoking related emphysema. Can you score emphysema somehow on CT scans ? is there a pre or post transplant moment where emphysema could be scored ? Not all smoker get COPD so this again is a bridge to far I would say. If this scoring is possible can it be related to the pack years ? (I hypothesis they do NOT correlate as this is obvious).

Response 1:  We really appreciate this comment. To evaluate a lung donor as suitable, especially in case of smoking history, the CT scan is useful as a first step as it is necessary to establish the presence of lung nodules, parenchymal thickening, anatomical abnormalities, atelectasis and especially emphysema. Unfortunately, in our study donor’s CT scan was not always available before the retrieval and so we did not utilize a precise score to calculate the presence of emphysema. For this reason the evaluation of lung’s emphysema was made directly on the lungs at the moment of the retrieval. The utilization of a CT score to better quantify the emphysema could be a necessary tool as suggested and for this reason we are collecting prospectively the donor's CT scan. However, we totally agree with the reviewer; also in our opinion there is not a direct correlation between the smoking pack-years and the extension of the emphysema as demonstrated by our study: we accepted in 23 cases lungs from donors with a smoking history >= 20 pack-years in which the extension of the emphysema was not a criteria to reject the organs. 

Point 2: smoking or pack years in a way is not the exclusion but the surgeon going for prelavation will during examination look (or feel) more for the emphysema and see if the deflate slower, bullae presence and so on. I know form our center that rejected lungs for emphysema can look very bad. And also that soem donor lungs used hold more pronounced emphysema get their pack years was not extraordinary. This discripance can you study this ? is their info from surgeon where they describe what they see and feel ?

Response 2:  Thank you for this reflection. This is a crucial point. In our experience of smoking-donors there was not always a direct correlation between the pack-year and the gravity of the emphysema. This finding allows us to retrieve in 23 cases lungs from heavy smokers donors deciding, after a careful evaluation, to implant them because they were suitable, without strong evidences of bullae or emphysema. It can be explained by the fact that emphysema is a multifactorial event not only caused by smoking.

Point 3: The biopsy study is very nice yet and biopsy does not reflect the emphysema and it is never homogenious. Could you quantify emphysema ? and also could you quantify the inflammation around the airways ? We know that donor exposure to pollution of smoking give inflammation of the small airway and that this impact on short term outcome, which you see. Can you link this ? LM would be cool the quantify and relate to the CT emphysema and FEV1 at 1 month and best FEV1.

Response 3:  We really thank the reviewer for this important food for though. The quantification of the different smoking lesions on the biopsies was performed by a dedicated pulmonary pathologist using a score a score system from 0 to 3 (0 = absence of lesion; 1 = involvement of less than 30% of parenchyma; 2 = involvement of 30-60% of parenchyma; 3 = involvement of more than 60% of parenchyma). As you can see from our pathological analysis, we graduated not only emphysema but also bronchiolitis, anthracosis and interstitial inflammation on biopsies; none of them resulted significantly different in donors with smoking history. It should be very interesting as you suggested, maybe in a next study, to find a model to correlate the CT scan findings, to FEV1 value and to histopathological findings.

Point 4: your FEV1 data show that the 20PY group have a lower FEV1 early on post LTx but that the other group catch up. Do you have 1 month data ? May it be that the early lower lung function results in the 20PY group makes that they have no reserve in the beginning causing the complication and longer ICU stay and hospital admission ?

Response 4: We thank the reviewer for this reflection. Unfortunately, we have 1 month-FEV1 only for the minority of patients and this does not allow us to drawn any significant conclusions.

Point 5: the outcome in the long term is not it seem. But are you misleaded (as all previous studies). The common feeling was that a lung from a smoker is inferior and could only be given to a COPD patients (as he does not deserve a top quality lung). Yet COPD patients in general do better in terms of survival. So could this bias the findings and be misleading ?

Response 5: We really appreciate your comment. As you can see from our results, survival and other mid and long-term major outcomes were not affected by smoking history even though the majority of patients transplanted with lungs retrieved from smoking-history donors were recipients affected by idiopathic pulmonary fibrosis that is the indication with the worst outcome and survival. Having in mind these considerations, we do not think that lungs from smoking donors could be reserved just for COPD patients. 

Round 2

Reviewer 2 Report

accept well done